# Pilgrimages on the Portuguese Way to Santiago de Compostela: Evolution and Motivations

Fátima Matos Silva [1,2,*], José Luis Braga [3,4], Miguel Pazos Otón [5] and Isabel Borges [1,3,6]

1. REMIT—Research on Economics, Management and Information Technologies, Department of Tourism, Heritage and Culture, Universidade Portucalense, 4200-072 Porto, Portugal; isabel.borges@iees.pt
2. CITCEM—Transdisciplinary Research Centre Culture, Space and Memory, Faculty of Arts and Humanities, University of Porto, Via Panorâmica s/n, 4150-564 Porto, Portugal
3. CIDI-IEES—Centre for Research, Development and Innovation-European Institute of Higher Studies of Fafe, 4820-909 Fafe, Portugal; jose.braga@iees.pt
4. CITUR—Centre for Research, Development and Innovation in Tourism, Polytechnic Institute of Leiria, 2411-901 Leiria, Portugal
5. ANTE—Territorial Analysis Research Group, Department of Geography, University of Santiago de Compostela (USC), 15705 Santiago de Compostela, Spain; miguel.pazos.oton@usc.es
6. CEGOT—Centre of Studies in Geography and Spatial Planning, Faculty of Arts and Humanities, Department of Geography and Tourism, University of Coimbra, 3004-531 Coimbra, Portugal
* Correspondence: mfms@upt.pt

**Abstract:** This research paper is based on the study of the evolution of pilgrimages on the Santiago Way, highlighting the Portuguese Way to Santiago—Central Portuguese Way and Coastal Portuguese Way—which has experienced massive popularity over the years. The primary objective of this work is to develop a comprehensive understanding of the pilgrims' motivations to undertake the Santiago Way pilgrimage. A mixed methods approach is adopted based on the simultaneous use of quantitative and qualitative data. So, an analysis of secondary data, provided by the *Oficina del Peregrino de la Catedral de Santiago de Compostela* and by the Municipal Department of Cultural Heritage Management of Porto is combined with a thematic analysis of seven interviews with stakeholders of the Portuguese Way to Santiago. The findings suggest that there is an increase in cultural and sports motivations, although spiritual and religious motivations continue to have a strong presence. The ecumenical character of the Santiago Way is also proved, given the large number of pilgrims of religions other than the Catholic one, who travel these paths—the vast territories that are traversed—until reaching the Cathedral of Santiago de Compostela. A new paradigm still needs to be registered, perceptible in the rise of *Turigrims*, pilgrims who benefit from support services that mitigate the hardships of the way.

**Keywords:** Portuguese Way to Santiago; pilgrimage; motivations; religious motivations; spirituality motivations; cultural motivations; ecumenism; places of worship; religious tourism

## 1. Introduction

The Santiago Way is a religious pilgrimage route with medieval origins that has become, especially in the last two decades, a religious, tourist, and cultural product with an enormous power of attraction.

It became a catalyst for the process of material construction of a pilgrimage route that has enabled the economic, social, and cultural sustainability of the different regions crossed by the different Paths. However, the aim of this article, and what motivated it, was the interest in knowing to what extent religious motivations, among many others that naturally exist, are still the basis that leads thousands of people, every year, to travel countless kilometres to reach Santiago de Compostela.

There are, undoubtedly, new types of pilgrims, with different motivations, stemming from different factors that interact with each other and lead to the concept of the pilgrim in

the most classic sense, but also as a new tourist who walks with the support of specialized companies—the "Turigrims"—or even a bicycle—the "bicygrim"—, among other cultural and/or religious tourists.

The analysis of statistical data from the *Pilgrim's Office of the Cathedral of Santiago* allows us to conclude that, in fact, religious motivations are still the driving force and also the strongest uniting factor among all types of pilgrims. However, nowadays, Santiago Way is characterized by being multi-motivational, not attending only to religious motivations (Amaro et al. 2018; Lois-González and Santos 2015).

Pilgrimage is an ancient form of mobility and a key precursor to modern tourism (Zapponi 2010). Tourism is one of the most relevant supports for the territory's sustainability, reaching a dimension that places it in a leading position among other economic activities. As it happens in many other tourist destinations, religious tourism is also one of the increasing segments in Portugal, as an economic activity and as an outcome of the movement of people. One route that has created greater dynamism, especially in the northern region of Portugal, is the Portuguese Way to Santiago, namely the Central Way and, more recently, the Coastal Way. These pilgrimage routes are the focus of this study.

Religious tourism is one of the oldest types of tourism in the world, being one of the first reasons people travel. The concept of religious tourism has been progressively transformed and updated. In fact, several studies indicate that religious tourism is a "fast-growing segment" of this industry (Griffin and Raj 2017). According to numerous definitions, religious tourism is related to "all travels outside the usual environment for religious purposes", which includes pilgrimage tours (Di Giovine and Elsner 2016, p. 722). Thus, religious tourism is a type of tourism, which includes people of faith who travel individually or in groups, for religious or spiritual purposes (Griffin and Raj 2017). The United Nations World Tourism Organization (UNWTO 1995) states that religious tourism can be one of the most effective instruments to provide broad and sustainable development.

The definition of religious tourism and pilgrimage tourism may seem alike, but there are, however, some differences (Vázquez de la Torre et al. 2010). Regarding religious tourism, the most important are the places of worship; in the case of pilgrimage tourism, in addition to considering the places of worship, the connection between places is important, with the need to travel to continue the visit, which contributes to the appearance of routes, itineraries or pilgrimage circuits (Gil de Arriba 2006), among which is the *Camino de Santiago*.

Religious tourism can be understood as an activity carried out by persons who travel for religious motivations or to attend events of a religious nature. It represents an opportunity for the development of tourist activities and for the economic and social development of the locations where they were built. Even more, because religious tourists are more loyal to the destination, they visit more than traditional tourists, as they repeatedly visit the same religious place in a shorter period (Robles 2001).

Religious and spiritual motivations are, usually, the basis of pilgrimages. However, core motivation is a form of transformation that includes inner values and questions about the meaning of life (Haab 1998). As Dyas (2020) says "And yet many people, including a large number who don't have any form of religious belief, still find such places a helpful way into exploring spirituality and enriching their lives. That is not an accident, because that is, in fact, what these places are for: to make us pause, reflect, and respond" (p. 7).

Spirituality is an individual phenomenon where one recognizes the importance of directing their lives towards something immaterial that is beyond or greater than themselves, with the recognition of some dependence on a higher power that is invisible or spiritual (Carlson and Martin 1999). Spirituality is an individual practice, but it is also related to the process of expanding beliefs around the meaning of life and connection with others. Hence spirituality is related to our inner consciousness, a particular form of energizing work action (Cavanagh et al. 2001; Guillory 2000).

Religiously motivated pilgrimages include belief in God (in a Christian sense) and the practice of traditional rituals. But as Dyas (2020) states "We are fortunate to have many ways of being a pilgrim today. The majority of pilgrims have some form of spiritual

connection with the Way" (p. 8). A pilgrimage with spiritual motives is based on the idea of an unstructured, individual, and transcendental relationship (Frey 2002). Even if, experienced individually, the pilgrimage is a social process that develops interactively over time. Although the majority of pilgrims come from a Christian background, many do not practice their religion in their daily lives, or on the Santiago Way.

Based on these assumptions, in this research paper, we intend to address the issue of religious tourism and pilgrimages to Santiago, studying the Portuguese Central Way and the Coastal Way between Porto and Valença.

We also intend to contextualise the Portuguese Way to Santiago both in historical and cultural terms, analysing statistical data from the Santiago Pilgrim's Office between 2004 and 2022 in various areas, and highlighting religious motivations. We will also analyse the statistical data relating to the five-year period 2018–2022 of the pilgrims who flocked to the Chapel of *Nossa Senhora das Verdades*—Support Centre for Pilgrims on the Portuguese Way to Santiago.

Furthermore, we will triangulate the statistical quantitative data with a thematic analysis of the indicators inherent in the transcripts of semi-structured interviews carried out with stakeholders of the Portuguese Way to Santiago.

This research paper is divided into seven sections. In addition to this introduction, the methods of data collection and analysis adopted will be described. On the other hand, a review of the relevant literature will be carried out, followed by the quantitative and qualitative analysis of the results, their discussion and, finally, in the conclusion, the limits of this study and future research lines will be highlighted.

## 2. Materials and Methods

Regarding methodology and in view of the elaboration of the historical and theoretical framework, a bibliographic collection and a critical review of the literature on religious and cultural tourism were carried out, with special emphasis on that which relates to the pilgrims' motivations. In this context, few studies were found.

Nevertheless, the empirical contribution of this research is based on a mixed methods approach (Ramseook-Munhurrun and Durbarry 2018). The combination of methods minimises the weaknesses of quantitative and qualitative methods applied in isolation and allows for a more comprehensive understanding of the area under study. Thus, triangulating methods allow multiple methods to be used to study a single research problem. A combination of qualitative and quantitative techniques contributes to more credible and reliable information (Phillimore and Goodson 2004). Triangulation limits personal and methodological bias and increases the fidelity of the study. Mixed methods have the following virtues (Ramseook-Munhurrun and Durbarry 2018, p. 118):

(a)   *provide a rich understanding of a phenomenon by combining exploratory, descriptive, and causal research designs;*
(b)   *address research questions better than single-method approaches; that is, a mixed-methods approach can simultaneously answer confirmatory and exploratory questions; and*
(c)   *allow development and justification of the conceptual model within one study.*

In fact, it is the research question inherent to this article—what are the push motivations that underlie the pilgrimages on the Portuguese roads to Santiago (Martínez-Roget et al. 2015)—that justifies the choice of secondary data analysis and interviews as the primary sources for this study. According to Marujo (2015), the "push" motives act as internal forces that persuade the individual to travel; they concern their internal and emotional sphere.

Thus, we chose to use secondary data that are useful for analysing long-term trends and allow hypotheses to be tested (e.g., there is a relationship between the *touristization* of the Portuguese Way to Santiago and an increase in non-religious motivation).

As De Haro et al. (2016, p. 92) state, "Interviews are used to collect data on motivations, attitudes, feelings, experiences, opinions, mental representations or life stories". Compared to a questionnaire, an interview is more flexible, has less possibility of standardisation, has greater subjectivity, has less structuring in the formulation of questions, and has a more careful choice of participants (De Haro et al. 2016).

The qualitative contribution of this research focuses on the collection and analysis of results from seven interviewees of relevant agents of the Portuguese Way to Santiago whose professional activity focuses on north-western Portugal. The participants were chosen for being experienced pilgrims and for having different connections to the Way, from innkeepers to members of Catholic organisations related to religious tourism to agents of municipal tourism organisations. The interview script contained a relatively small number of key questions, allowing the use of queries to clarify certain topics and deepen certain themes raised by the interviewees (Finn et al. 2000).

The interview script contained a relatively small number of key questions, allowing for the use of probes to clarify certain topics and elaborate on certain themes raised by interviewees (Finn et al. 2000). The people interviewed all have an intense relationship with the Portuguese Way of Saint James. In their capacity as hostel owners, academics, restorers, or simply friends of the Camiño, they carry out their professional activity and their leisure time closely linked to the phenomenon of pilgrimages to Santiago de Compostela. Their different professions and occupations constitute a way of approaching the knowledge of the pilgrims' reality from different points of view, based on the performance of their professional tasks. But in addition, each of these people dedicates a good part of their free time to teaching, disseminating, and promoting the Way of Saint James, moved by a vocational force. After any given year, each of these people meets and talks with hundreds and hundreds of pilgrims, which is a kind of continuous participant observation. These people interviewed are, therefore, "intermediaries" between the pilgrims and the researchers since they transmit their experiences, perceptions, and impressions, and code them according to their own professional activity.

The analysis of the interviews followed a thematic analysis (Bryman 2012, p. 580), in which a theme refers to:

(a)   a category identified by the analyst through his/her data;
(b)   that relates to his/her research focus (and quite possibly the research questions);
(c)   that builds on codes identified in transcripts and/or field notes; and
(d)   that provides the researcher with the basis for a theoretical understanding of his or her data that can make a theoretical contribution to the literature relating to the research focus.

In parallel, the authors developed the analytical study, converting into tables the official statistics obtained by the Pilgrim's Office of the Cathedral of Santiago de Compostela. This study also presents statistical data collected from 2018 (date of opening of the Our Lady of Truths (*Nossa Senhora das Verdades* Chapel, Pilgrim Support Centre/Portuguese Way to Santiago in Porto) until 2022. Regarding the data presented here, there are certain constraints: (a) in the year 2020, the centre was closed from 18 April until the end of May, due to COVID-19; (b) in late September 2020, rehabilitation works were carried out on the building until 13 October; (c) on 15 January 2021 until 15 April, the chapel was closed due to COVID-19.

## 3. The Portuguese Way to Santiago: The Evolution of Pilgrimages

Pilgrimages are journeys made, mainly, for religious, cultural, or spiritual reasons. Ian Reader's "Pilgrimage in the Marketplace" (Reader 2014) also includes politics and/or commerce, showing that pilgrimage operates in and through the marketplace via the deployment of consumer activity, publicity, and promotion, usually involving visits to places considered sacred or important to the tradition of a particular religion or culture. They have been a common practice throughout human history, in numerous cultures and religions.

The pilgrimage to Santiago de Compostela is one of the oldest Christian pilgrimages, captivating thousands of people every year. The destination is the Cathedral of Santiago de Compostela, in Galicia, Spain, where the tomb of the apostle James the Great is located (López Gómez 2010).

The Way of Santiago was developed around the twelve century when the remains of St. James or Santiago were probably found in Santiago de Compostela (Pombo 2007). Over the centuries, it has increased as well as declined in popularity (Moreno 1992). In recent decades its growth has been exponential, continuing to be seen as a religious path, but also as a spiritual and cultural route. In 1987, it was acknowledged as the First European Cultural Itinerary.

In each Holy Year or Jacobean/Xacobean Year[1], as it is called in Galicia, there is a remarkable increase of pilgrims attracted to this religious and cultural itinerary (Balasch Blanch and Arranz 2013).

The various paths (Figure 1) are generally signposted by directional signalling in yellow arrows, painted everywhere, or by various types of scallop shells produced in diverse materials, as a result of various funding programmes.

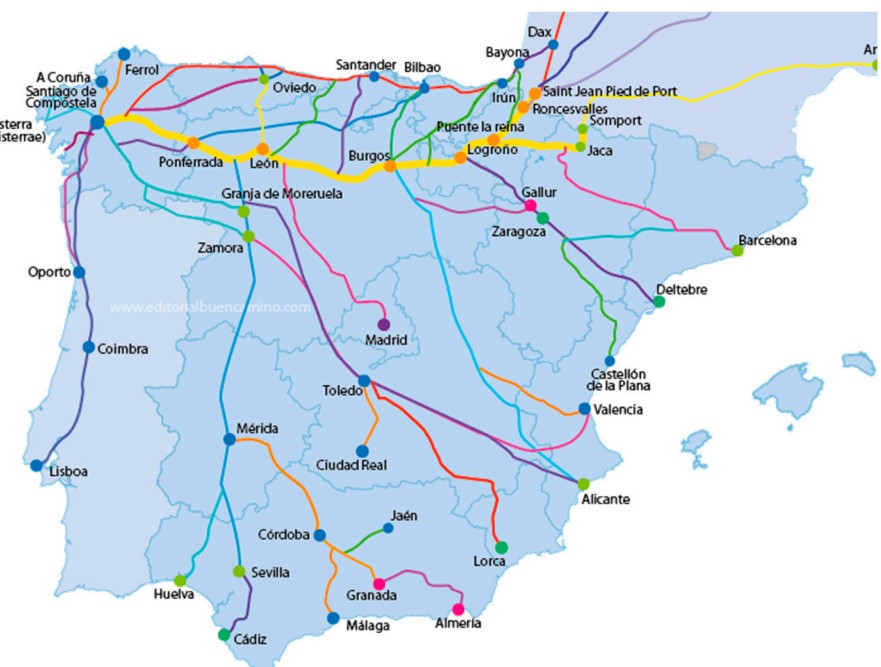

**Figure 1.** Map of the Ways to Santiago in Southwest Europe. Source: Mapa—Camino de Santiago. Source: Mapa—Camino de Santiago. Guía definitiva: etapas, albergues, rutas (editorialbuencamino. com, accessed on 20 January 2023).

The Portuguese Way is divided, among others, into the Coastal Portuguese Way, the Interior Portuguese Way, and the Central Portuguese Way (Figure 2). The last one is the most travelled route. Beginning in Lisbon, passing through Coimbra, Porto, and ending in the Portuguese territory at Valença do Minho.

It is worth mentioning the importance of the Central Way between Porto/Vila do Conde/Póvoa de Varzim/Barcelos/Ponte de Lima/Paredes de Coura/Valença where other routes merge, reinforcing this path as the most important of the Portuguese roads to Santiago. This route has been in great demand in recent years and is widely seen as being the main Portuguese Medieval or Central Way.

The Central Way starts primarily in Porto and is about 127 km long in Portuguese territory. It is generally a rather rough stretch, making it a rather difficult route to undertake on foot (Silva and Borges 2018).

The Coastal Way, which is 149.5 km long, is a path that starts in Porto and passes through the current municipalities of Matosinhos, Maia, Vila do Conde, Póvoa de Varzim, Esposende, Viana do Castelo, Caminha, Vila Nova de Cerveira and Valença. It is a Way that, until it reaches Galicia, always follows along the coast and the banks of the river Minho. This route is much flatter than the stretch of the Central Portuguese Way between Porto and Valença. As we will verify, it is one of the Ways of Santiago that has been rising in recent years.

There are no consistent statistics regarding the thousands of tourists and visitors, pilgrims or not, who annually travel to Santiago de Compostela by means of motorised transport—bus, car, plane, or train.

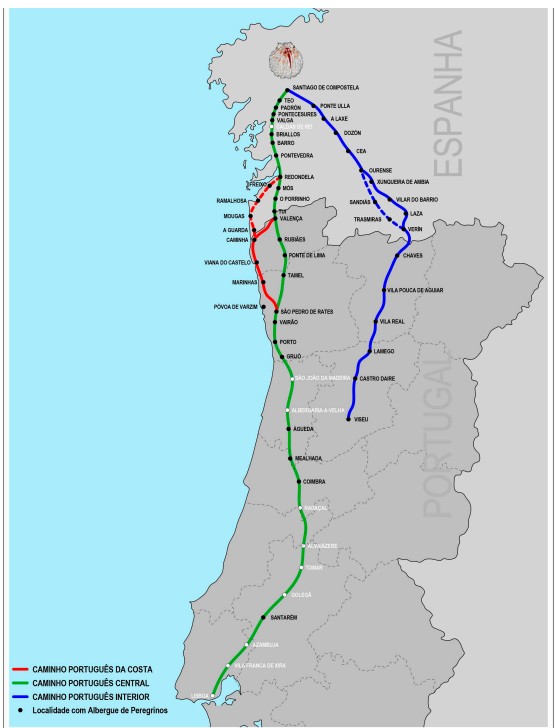

**Figure 2.** Portuguese Way to Santiago. Source: https://caminhosantiagoviana.pt/ (accessed on 20 January 2023).

The data analysed here exclude the vastness of the population that visits Santiago and its various cultural, religious, and other facilities that characterise the cultural heritage associated with the Ways of Santiago. The statistical data provided by the *Oficina del Peregrino* of the Cathedral only refer to the pilgrims who go there to receive the Compostela, which corresponds to only about a third of the total (Oficina del Peregrino de Santiago de Compostela n.d.).

In the celebrations of each Holy Year, as we can see in Table 1, the number of pilgrims, who have registered in the *Oficina del Peregrino*, has exceeded all expectations. Nevertheless, after the 2010 Holy Year, the growth of pilgrims has been very considerable, and in 2016 (not a Holy Year) exceeded the number of pilgrims who travelled the several routes during Holy Years.

During 2022, the number of pilgrims who arrived in Santiago was even greater than in 2019, the year in which it had reached the highest figures. Given the reduction in the restrictions related to the pandemic by COVID-19, the Xacobean Holy Year, which took place in 2021, was extended to 2022, enabling more pilgrims to enjoy the event already anticipated for 11 years.

**Table 1.** Evolution of pilgrims in general, pilgrims from the Portuguese Coastal and Central Ways from 2004 to 2022. Source of data: own elaboration from Pilgrim's Office statistics.

| Years | Pilgrims All The Ways | Central Way | Coastal Way | Portuguese Ways-Totals |
|---|---|---|---|---|
| **Holy Year 2004** | **179,944** | **15,839** | **-** | **15.839** |
| 2005 | 93,924 | 5507 | - | 5.507 |
| 2006 | 100,377 | 6467 | - | 6.467 |
| 2007 | 114,026 | 8110 | - | 8.110 |
| 2008 | 125,141 | 9770 | - | 9.770 |
| 2009 | 145,877 | 11,956 | - | 11.956 |
| **Holy Year 2010** | **272,135** | **34,147** | **-** | **34.147** |
| 2011 | 183,366 | 22,062 | 41 * | 22.062 |
| 2012 | 192,488 | 25,628 | 177 * | 25.628 |
| 2013 | 216,880 | 29,550 | 274 * | 29.550 |
| 2014 | 237,983 | 35,501 | 268 * | 35.501 |
| 2015 | 262,516 | 43,151 | 399 * | 43.151 |
| 2016 | 277,854 | 49,538 | 2600 | 52.138 |
| 2017 | 301,036 | 59,233 | 7329 | 66.562 |
| 2018 | 327,374 | 67,820 | 13,839 | 81.659 |
| 2019 | 347,578 | 72,357 | 22,292 | 94.649 |
| 2020 | 54,134 | 10,252 | 2736 | 12,915 |
| **Holy Year 2021** | **178,912** | **32,315** | **7813** | **40,128** |
| **Holy Year 2022** | **438,000** | **93,193** | **30,609** | **123,802** |

Data marked with (*) correspond to starting points recorded by the authors, which geographically correspond to the Portuguese Coastal Way.

Pilgrims from all over the world have walked the dozens of routes that make up the Way, enjoyed the experience and have given back to the Way of Santiago its usual flow, reaching the number of 438,000 pilgrims, which makes 2022 the most successful Holy Year in the history of pilgrimages. Although the Way of Santiago is carried out on any day of any month, there are certain months of the year that continue to receive more pilgrims than others. Specifically, August continues to be the month with the highest number of walkers, with more than 85,000 people in 2022.

As far as the Central Portuguese Way to Santiago is concerned, and focusing on the analysis only of recent years, 2019 also exceeded all expectations. Thus, the Central Way, up to the end of December 2019, had been travelled by 72,357 pilgrims, to which must be added the figures of those who pilgrimaged along the Coastal Way (22,292), making a total of 94,640. Affected by the years of the COVID-19 pandemic, especially in 2020, it recovered in 2021, the Holy Year, exceeding all expectations in 2022, the second Holy Year, having been covered by 123,802 pilgrims.

Regarding the Portuguese Coastal Way, we have managed to count some pilgrims by their geographical place of departure since 2011 (Table 1) but there is only the registration, at the Pilgrims' Office, since 2016 (2600 pilgrims). The number of pilgrims tripled in 2017 (7329), continuing to grow exponentially in 2018, reaching 13,839 pilgrims in 2022, almost six times more than the initial figures.

Portugal represents, at this moment, the second most important source of pilgrims. Even so, most of the walkers are not Portuguese (Figure 3). The number of pilgrims of other nationalities has been growing. Since 2017, Portuguese pilgrims have had a slight reduction compared to previous years, increasing again in 2018.

In 2022, as far as itineraries are concerned, the French Way was travelled the most by pilgrims, with 226,887, followed, as has been usual in recent years, by the Portuguese Way, with 93,193 pilgrims. Its variation closer to the coast—the Portuguese Coastal Way—grew once again, having been covered by 30,609 pilgrims, managing to surpass other routes, such as the English Way, with 24,205 pilgrims, thus placing it as the third most chosen option by pilgrims.

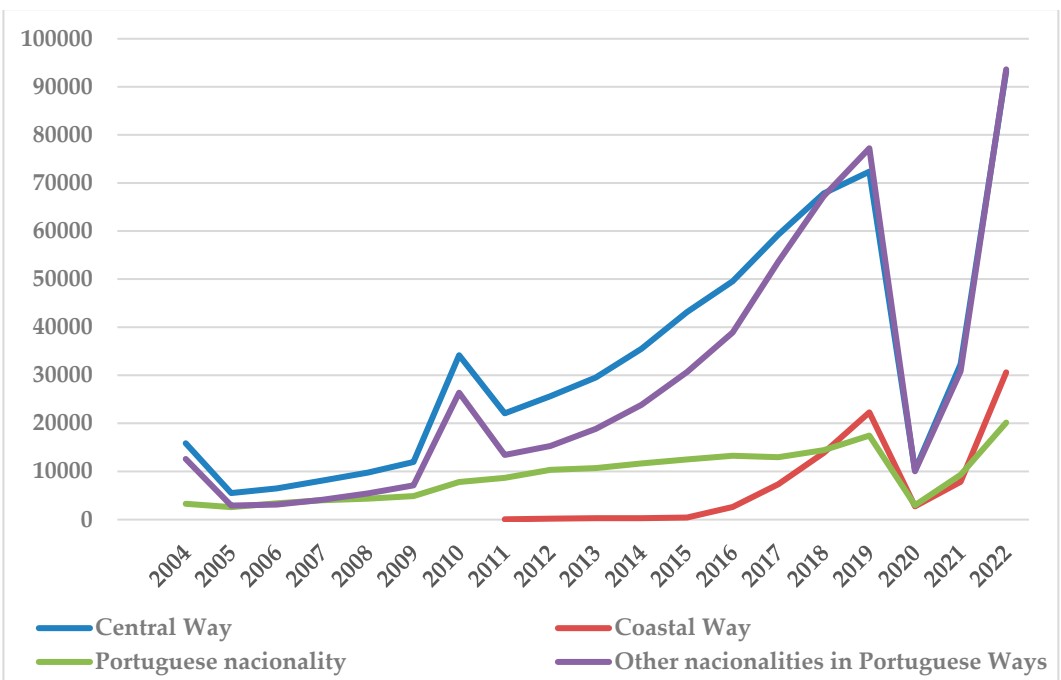

**Figure 3.** Evolution of the number of pilgrims on the Portuguese Coastal and Central Ways and nationalities (Portuguese and others) between 2004 and 2022. Source: own elaboration from Pilgrim's Office statistics.

Evidently, less travelled paths were also attracting people after 2022, especially the Primitive Way, with 21,360 pilgrims, and the Northern Way, with 20,866 pilgrims.

Regarding the nationalities of the pilgrims, the Way of Santiago is travelled every year by hundreds of thousands of people who come from all over the world. The Spanish nationality is still the most important among pilgrims, with a total of 239,417 in 2022. The United States was the third most represented country, with a total of 26,000 pilgrims. In second place is Italy with 27,078. These countries are followed by Germany and Portugal, with 23,212 and 20,166 pilgrims, respectively.

In the last years, the number of women pilgrims also continues to be higher than that of men. The year 2022 continues to have a very similar percentage of participants by gender. Specifically, a total of 231,461 women, 206,860 men, and 2 persons who did not wish to specify travelled the different paths.

The Way is typically completed on foot. However, it may also be completed by other means recognised as true forms of pilgrimage, such as by bicycle or on horseback. Table 2 shows that the most commonly used means of transport are on foot and bicycle, although there is also some mention of horse and wheelchair transport. There is no information about the means of transport used in each of the itineraries.

In the year 2022, a total of 414,340 pilgrims walked the Way. The second most frequent mode of transport was the bicycle, with a total of 22,863 pilgrims, followed by the horse with 545, and 127 completed it in wheelchairs. The number of "*bicygrims*" has been increasing, which indicates that this way of travelling the different Jacobean routes is becoming more and more common. In the last few years, it has also been considered the route was completed by sea/river, with 448 people.

As already mentioned, the Portuguese Coastal Way has experienced very significant growth in recent years. Given that the city has also had a very significant increase in tourist demand due to several factors, we consider that there is a very close reason for the increase of pilgrims who depart from this city, especially after 2014.

**Table 2.** Evolution of pilgrims according to the type of transport. Source of data: own elaboration from Pilgrim's Office statistics.

| Year | On Foot | Bicycle | Horse | Wheelchair |
|---|---|---|---|---|
| Holy Year 2004 | 156,952 | 21,260 | 1672 | 60 |
| 2005 | 76,674 | 16,985 | 242 | 23 |
| 2006 | 81,783 | 18,289 | 294 | 11 |
| 2007 | 93,953 | 19,702 | 364 | 7 |
| 2008 | 103,669 | 21,143 | 290 | 39 |
| 2009 | 120,605 | 24,892 | 341 | 39 |
| Holy Year 2010 | 237,852 | 32,926 | 1315 | 42 |
| 2011 | 153,065 | 29,949 | 341 | 11 |
| 2012 | 164,778 | 27,407 | 281 | 22 |
| 2013 | 188,191 | 26,646 | 977 | 66 |
| 2014 | 211,033 | 25,332 | 1520 | 98 |
| 2015 | 236,773 | 25,346 | 326 | 71 |
| 2016 | 254,025 | 23,347 | 342 | 125 |
| 2017 | 278,490 | 21,933 | 417 | 43 |
| 2018 | 305,655 | 20,774 | 589 | 209 |
| 2019 | 327,281 | 19,563 | 406 | 85 |
| 2020 | 49,557 | 4493 | 59 | 12 |
| Holy Year 2021 | 158,833 | 10,285 | 167 | 34 |
| Holy Year 2022 | 414,340 | 22,863 | 545 | 127 |

The data highlight the idea that the different Santiago Ways have been one of the engines of the economic growth of several regions. The existence of a Santiago route in the territory of a municipality, in any region, increases religious and cultural tourism, but it also increases their obligation to create sustainable conditions for pilgrims and residents.

## 4. Pilgrimage Motivations

The Way of Santiago is a physical journey that is made with effort, stage by stage. But it is also a spiritual journey, among other reasons, full of teachings and learning, where each person lives the Way in a different manner.

In the Middle Ages and in later times, devotees from all social classes, including royalty, peregrinate almost exclusively for religious reasons to Santiago. Nowadays, we find a considerable percentage of pilgrims and walkers who do not do it only for religious reasons, but for cultural or other motivations.

Although the pilgrims' motivations may vary, we believe that religious motivations, as proven by the statistical data, are still the main ones today.

The motivations of pilgrims to undertake the Santiago Way vary greatly among individuals. Many influences, such as cultural ones, cause people to undertake pilgrimages, and there are as many motives for pilgrimage as there are spiritual or religious needs.

Motivation is "something which commits people to a course of action, i.e., the driving force which exists in all individuals" (Raj et al. 2015, p. 109). Studies have identified different motives for religious tourism and for pilgrimages. Historically, a pilgrim was described as a person who walked to a holy place for religious motives (Rinschede 1992), a traditional pilgrimage conducted with a strong religious motivation (Shinde 2007). But, nowadays, the modern pilgrim is not necessarily motivated by religious reasons (Štefko et al. 2015) and travels for many other reasons (Oviedo et al. 2014).

Frey (1998) conducted an anthropological analysis based on the increase in the number of pilgrims since the 1980s. The author stated that the current pilgrims went to Santiago for several reasons that might change throughout the Way. She differentiated between (1) religious motives (such as the fulfilment of promises, a crisis, the renewal of faith, or praying for others), (2) spiritual motives (personal searches or inner journeys of transformation), and (3) historical and cultural motives combined. Frey, being a pilgrim of the Way, claimed

that pilgrims noted the difference between religious and spiritual motives as well as the difference between orthodoxy and personal devotion.

According to Francisco Singul (1999), there are five main motivations for walking the Santiago Ways: (i) traditional religious (devotion); (ii) cultural (medieval art, history); (iii) ecological (contemplation and enjoyment of the landscape and the natural environment); (iv) spiritual and ecumenic; and (v) personal (meditation on one's life).

For some pilgrims, walking along the Way is a way of getting to know the local culture and learning more about the history and traditions of the region. The various routes pass through several cities and towns with a vast and diverse cultural and historical heritage, which is an important tourist-cultural attraction. Walking along the Way is exciting and challenging for some pilgrims. The routes offer walkers the opportunity to experience different landscapes, terrains, and climates as well as meet people from all over the world.

For Oviedo et al. (2014), walking to Santiago is a way of finding peace, spiritual tranquillity, and spiritual growth. The journey can offer an opportunity for reflection and meditation, time to reflect on possible decision-making and changes in way of life and motivations that have gained ground relative to religious motivations. In fact, many pilgrims walk as a way of challenging themselves and testing their physical and mental limits or overcoming a personal crisis, such as a divorce or the death of a loved one.

The data provided by the Pilgrim's Office compiles only three types of motives: religious, non-religious, and religious and other unspecified. However, they are data that reflect the totality of pilgrims on the different Jacobean routes. We admit that it is possible that these general figures also characterise the motivations of the pilgrims who travelled the Portuguese Way to Santiago in the variants under study.

Analysing the data in the graph (Figure 4), we can quickly conclude that the pilgrimages made for religious or other unspecified motivations between 2004 and 2022, even considered separately, largely exceeded the number of pilgrims who made the Way without religious motivations. However, it is the type "religious or other reasons" that exceeds the other types throughout the 19 years that the statistics record, with two exceptions. These exceptions concern the Holy Years 2004 and 2010, in which the pilgrims' motivations for religious reasons largely exceeded the other comprehensive type of "religious or other motivations", especially in 2004.

In the last two Holy Years—2021 and 2022—there has been a paradigm shift whose motives are hard to find, so even though religious motivation is largely predominant, the overarching type "religious or other motivations" has surpassed stated religious motivations as the sole motive.

It would be necessary for the statistics made available by the Pilgrim's Office to be more detailed in terms of motivations in order to be able to accurately assess what has justified this change of paradigm.

One hypothetical justification has to do with the worldwide publicity that has been given to the Ways of Santiago, which has enabled a significant increase in the number of pilgrims, with different motivations, namely cultural. It also seems to us that this is a consequence of the post-pandemic phase of the COVID-19 pandemic. The COVID-19 pandemic has undoubtedly had an impact on the motivations of pilgrims undertaking the Santiago Way. The contemporary world was unaware of the pandemic situation and had never been restricted from going out, visiting, experiencing new situations, or contacting nature and oneself—in a word, going on a pilgrimage. The uncertainty of the future was distressing. When they had this chance, as well as the recovery of tourism at the world level, the number of pilgrims also increased exponentially, although keeping religious motivations as the priority, others also joined in.

Oviedo et al. (2014) state that, considering the complexity of the pilgrims' motivations, individuals with "various, often contrasting, motivations and expectations walk side by side" (p. 433) on the Santiago Way pilgrimage route.

There is no shortage of reasons to do the Santiago Way, even if it is a challenge for many pilgrims. Whatever the main motivation is, they are all valid because it is a unique

experience. As the journey ends, they are not the same person as at the beginning of the pilgrimage.

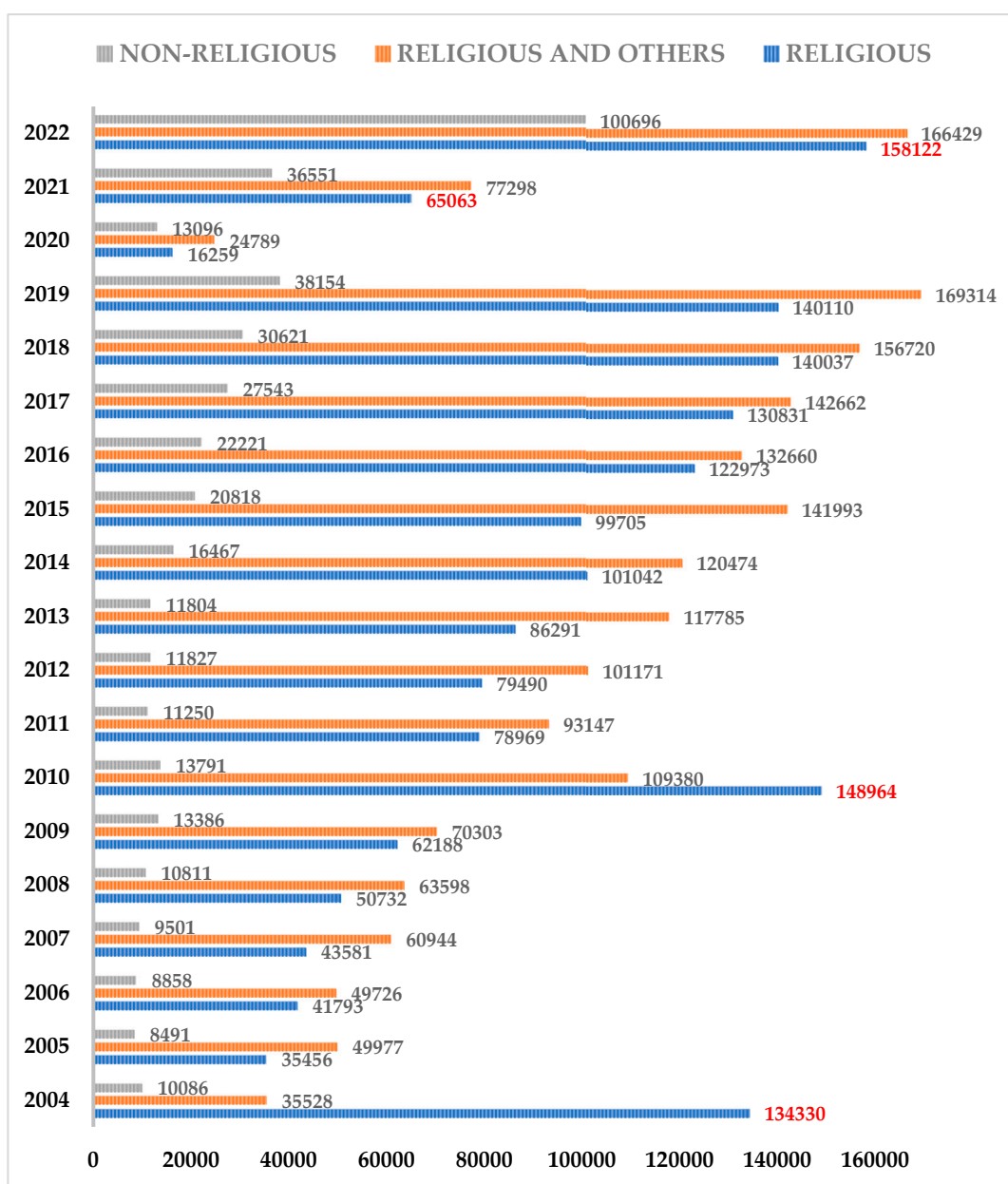

**Figure 4.** Evolution of pilgrims according to motivations. Source of data: own elaboration from Pilgrim's Office statistics.

## 5. Driving Factors for Pilgrimage Mentioned by the Interviewees

In an early phase of the *touristization* of the way, from 1993 onwards, one of the interviewees asserts that most pilgrims had religious motivations, namely those associated with the Catholic faith. According to the same participant, nowadays, more than religious factors, what is at the origin of the pilgrimage is the spiritual dimension.

The search for something higher, something inherent to the human being, the discovery of a sign, of self-knowledge, of an answer, is where this spirituality is based, considering that spirituality is a form of contemporary religiosity. This quest may be triggered by diverse factors: individuals (young or not) who are going through a sentimental problem, people who want to change professions, or even individuals who have lost loved ones. In fact, the testimonies of the interviewees corroborate the need to overcome, as already mentioned.

In turn, another interviewee, co-owner of a restaurant where many pilgrims pass, claims that the motivations are divided "half for religious reasons, half for spiritual reasons". Another interviewee, associated with a Catholic religious organisation, claims that the vast majority of pilgrims travel for a "cultural and religious motivation" and a minority for sports. An interviewee belonging to the Department of Culture and Tourism of a municipality through which one of the Portuguese itineraries passes through is of the same opinion: "There is also a sports motivation, and this is clearly also on the rise".

There is, therefore, a tendency for the sports practice and the spiritualities associated with the Way to increase as driving factors of the pilgrimage. Cultural motivation comprises spiritual motivations and is distinguished from "a purely religious dimension of faith". This cultural motivation may also have been triggered by the expansion of low-cost airlines which proliferate at the Francisco Sá Carneiro airport, which serves Porto, but also the whole northern region of Portugal and Galicia in Spain. The dimension of the various spiritualities is an individualist demand that contrasts with a collective vision of pilgrimage.

The pilgrims who move for religious reasons are not necessarily Catholic. The testimonies of the interviewees reveal that the Way is ecumenical, as one interviewee, associated with a Catholic organisation and an experienced pilgrim, says: "We actually find people of all nationalities and some more and people of different religions walking the Way of Santiago".

According to the representative of an institution devoted to the study of the Ways, "human nature has in its nature the desire to go on a pilgrimage", according to this interviewee "it has to do with our desire for transcendence". It follows that pilgrimage is an inherent dimension of the monotheistic religions (and not only...). Muslim, Jewish, Orthodox, and Protestant believers often make the journey, as well as Christian minorities from China and South Korea, or Japanese of monotheistic religion.

One participant in this study, co-owner of a restaurant located near a pilgrimage route, even has "half a dozen scallops without the cross because the Muslims come and don't want a scallop with a cross on it". And as for the Jews, he mentions: "They pass by, they even pass by with the kippah". Another interviewee, vice-president of a Catholic institution, is also emphatic in this regard when he states: "I have never done the Pilgrims' Way to Santiago with people of other religions, but I am perfectly aware that there are Buddhists, Muslims, and Hindus doing the Way to Santiago".

Still, within the Christian confession, the Orthodox and Lutherans also frequent the Way regularly. Otherwise, let us pay attention to what the representative of an institution specialising in the Portuguese Way of Santiago says: "Since 2005, Orthodox, but they were Ukrainian Orthodox, came to us and for many years the director of this pilgrimage was Doctor Iuri, professor at our Physics Faculty here". In what concerns Protestants, he adds: "There are Lutherans and there are those of Dr Pina Cabral, who was an Anglican bishop (...). He was a bishop and very interested in the ways".

An interviewee belonging to the hierarchy of a Catholic religious organisation states that he made the Way of Santiago with agnostics. He says: "I've met, I've crossed paths and I've already made the road to Santiago with people, therefore, agnostics. People without any religion. I have never walked the Camino de Santiago with people of other religions, but I am perfectly aware that there are Buddhists, there are Muslims, there are Hindus doing the one in Santiago. That I don't have the slightest doubt. As there are Catholics who will make Buddhist paths and go to India to do some paths of pilgrimage".

On the other hand, an employee of an Interactive Tourism Shop in a city located near the city of Porto refers to the existence of "secular pilgrimages". According to him, "This goes against the spirit a little bit, not to say that it goes against the spirit a lot, but it already exists". In fact, even among the hosts of the pilgrim hostels, there is the awareness that associating the Way with the Christian religion may drive away "many people that hate the Church, but that is on the Way". According to an official of an association of hostels of the Portuguese Way, the good reception of pilgrims discourages "attaching too much [the Way] to the image of the Church".

The quest on the Way of Santiago, according to the same interviewee, lies more in the journey itself and less in the arrival at the shrine, as happens in Fátima. For the pilgrim to Santiago, "the pleasure is to go, not to arrive", according to this interviewee.

In turn, the interviewee responsible for a Catholic organisation associates the increase in the number of pilgrims on the Portuguese Way, which is clearly visible in the charts and tables mentioned above, with the "overbooking of the French Way".

Since the Compostela is an indulgence, even nowadays the Way may be seen as an alternative to imprisonment due to a crime committed in the pilgrims' region of origin (Shaver-Crandell 1982). The same participant witnessed this phenomenon when he came across a group of inmates from Ourense who were on their way to Santiago as part of "their end-of-year project (...) an activity to make the Way of Santiago". The inmates were travelling during the Holy Year, which according to the same interviewee is a time of "full forgiveness of our sins". In this sense, the pilgrimage is a reaction "to the materialism of the world".

## 6. The Rehabilitation and Evolution of the Portuguese Coastal Way

In recent years, Portuguese regions have been developing efforts to include several Jacobean routes in their territories, promoting their certification, thus allowing the respective municipalities to stand out in the Portuguese contemporary tourism framework. However, the responsibility of these areas and the respective agents of the territory, namely the municipalities, is increased since they benefit from the potential that the routes offer them, but also require from them. Unfortunately, some of these routes do not have conditions to receive pilgrims, and others have no historical justification for being considered Jacobean routes.

The pilgrims, currently in large increasing numbers, as we mentioned, help to improve the economy of the territories they pass through, also contributing to greater visibility and notoriety of those places. Dissemination is carried out through testimonials or sharing, both in social networks and through the relationship with other pilgrims and with the communities themselves.

They also promote and disseminate not only the routes to Santiago, but also the cultural heritage and, above all, the built religious heritage. Chapels and churches associated with the Portuguese Way have been systematically subject to rehabilitation, restoration, conservation, and enhancement projects.

The most emblematic case is the Portuguese Coastal Way. This Way started to be walked in the 18th century, getting lost in time, but in recent decades, its memory has been forgotten. However, it has recently been rehabilitated, significantly increasing the number of pilgrims that walk it (as we mentioned and as we see in Table 1), after an intense campaign of promotion and the creation of accessibility in its various aspects, because of the union of ten municipalities from Porto to Valença, passing through Matosinhos, Maia, Vila do Conde, Póvoa de Varzim, Esposende, Viana do Castelo, Caminha and Vila Nova de Cerveira. As an integral part of the valorisation project, new signage equipment—directional and informative—was also implemented along the path in order to make it more uniform and fluent in its signalling.

The Portuguese Coastal Way is 149.5 km long and has 462 heritage monuments present in the 10 municipalities through which it passes. The 51 places of religious worship, churches, and chapels that we studied in terms of accessibility have been improved. At the beginning of the path, in Porto, the *Nossa Senhora das Verdades* Chapel was adapted to become the Interpretation Centre for Pilgrims of the Coastal Way. The creation of this interpretation centre helped to rehabilitate and give new use to a ruined architectural structure, now with new functionality and usefulness. In the chart in Figure 5, we can observe the evolution in the number of pilgrims according to the chosen variant of the Portuguese Way.

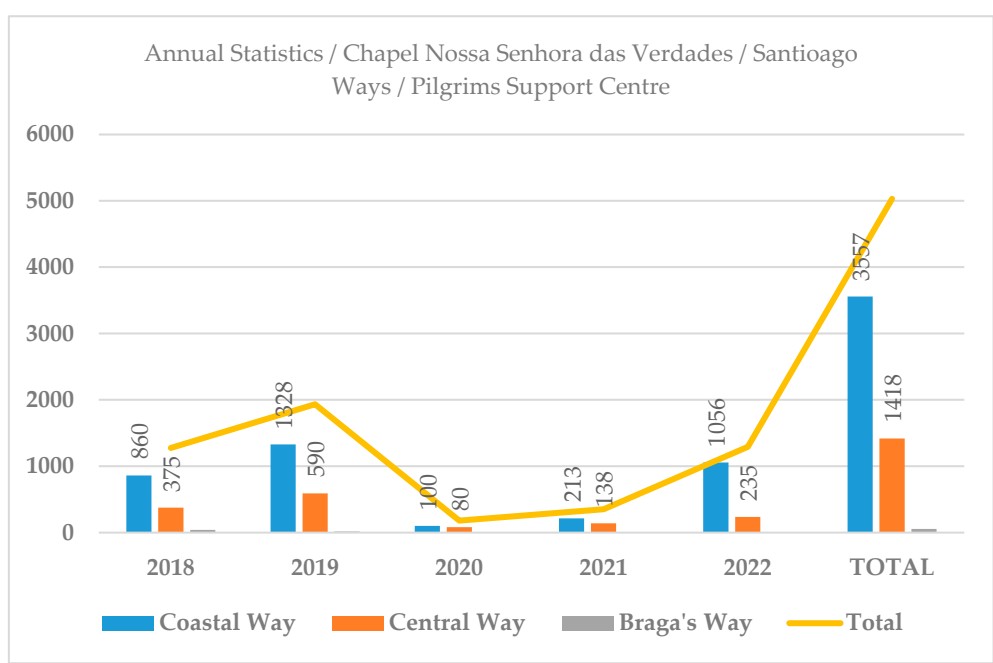

**Figure 5.** Evolution of the number of pilgrims according to the chosen route variations from 2018 to 2022. Source of data: Municipal Department of Cultural Heritage Management of Porto.

From the analysis of the chart in Figure 5, we can infer that the steady increase in the number of pilgrims between 2018–2022 was only interrupted by COVID-19. When the effects of the pandemic began to be overcome, the influx of tourists registered an exponential increase. There is also a predominance of the Coastal Way in all the years of the interval. These numbers differ from the statistics presented in Table 1. The demand for the Coastal Way is based on the presence of the sea, which is probably one of the pull factors for a pilgrim who does not have a purely religious motivation but a spiritual or even non-religious one. By "pull", we refer to the attractiveness of the destination (Martínez-Roget et al. 2015; Marujo 2015).

## 7. Conclusions

The present study sought to answer the research problem of knowing whether religious motivations still constitute the main push factors of pilgrims on the Santiago Way. The analysis of the secondary data from the Pilgrim's Office made it possible to highlight the predominance of purely religious motivations. Although there is a prevalence of religion, there is a clear tendency for motivations of a cultural, spiritual, and sports nature to gain relevance. This same perception was expressed in the testimonies of the seven stakeholders of the Portuguese Way who were the subject of the interview.

The result of the interviews also underlined the ecumenical role of the Portuguese Way of Santiago, travelled by many different pilgrims of different religions whose motivation is, above all, a spiritual character.

The emergence of "turigrims" and "bicygrims", detected in the extant literature, is supported by the quantitative and qualitative data collected. Nevertheless, religious motivations still constitute the main motive of the Jacobean pilgrimages originating in Portugal.

This study has some limitations. Firstly, the fact that many pilgrims who go to Santiago do not go to the Pilgrim's Office to receive the Compostela, which increases the authenticity of the sample. This circumstance may also influence the data collected at the *Verdades* Chapel. In relation to the qualitative data, more than 10 h of interview recordings were garnered. All of them have given rise to extensive research material which, without a doubt, will allow this investigation to continue in subsequent articles.

In any case, the qualitative data present some novelties regarding the ecumenical nature of the way and even the emergence of lay pilgrimages.

Another noteworthy aspect is the fact that the data from the Pilgrim's Office and the *Verdades* Chapel do not coincide regarding the variants of the route chosen by the pilgrims. Here we can see a preponderance of the Coastal Way, which contradicts the data from the Pilgrim's Office, where the Central Way is hegemonic.

Future research could try to explain this apparent incongruence and find out how statistics are being collected and how they can be more faithful to the motivations of the universe of pilgrims who come to Santiago de Compostela.

**Author Contributions:** Conceptualization, F.M.S., J.L.B., M.P.O. and I.B.; methodology, F.M.S. and J.L.B.; formal analysis, F.M.S., J.L.B., M.P.O. and I.B.; investigation, F.M.S., J.L.B., M.P.O. and I.B.; data curation, F.M.S. and J.L.B.; writing—original draft preparation, F.M.S., J.L.B., M.P.O. and I.B.; writing—review and editing, F.M.S., J.L.B., M.P.O. and I.B. All authors have read and agreed to the published version of the manuscript.

**Funding:** This research received no external funding.

**Institutional Review Board Statement:** Not applicable.

**Informed Consent Statement:** Not applicable.

**Data Availability Statement:** Not applicable.

**Conflicts of Interest:** The authors declare no conflict of interest. The citizens interviewed have given their consent for their opinions to be made public, however, we have chosen to preserve their identity or to specify which institution they belong to.

## Note

[1] Holy Years occur when the 25th of July (Santiago's Day), corresponds with a Sunday. The year 2021 was a Jacobean/Xacobean Year, however, given the Covid-19 pandemic, Pope Francis has also declared 2022 a Holy Year.

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
