# Peer review of "Pilgrimages on the Portuguese Way to Santiago de Compostela: Evolution and Motivations"

_religions, doi:10.3390/rel14081017_

Round 1

Reviewer 1 Report

This is a promising submission but in my opinion it can be improved by engaging with the comments I have added to the text. 

I have entered a number of edits in the text.

Author Response

We appreciate the reviewer suggestions and opinions which, of course, 
contribute to the improvement of our article, so we have taken them into account. 
We thank the reviewer for the time and effort.
The reviewer was kind enough to make the comments and suggestions directly 
in the text. However, given the difficulty of mentioning each of them here, we have chosen to make the changes also directly in the text. 
It can be noted that all comments and suggestions have been taken into account, definitely contributing to an improvement of the article.
Thank you.

Reviewer 2 Report

I have attached my comments and suggestions.

The use of English in various instances appears rather awkward, making it difficult to understand the text smoothly. 

Author Response

We appreciate the reviewer proposals and opinions which, of course, contribute to the improvement of our article, so we have taken them into account. 

Changes were made to the manuscript and explanations were given in the attached document.

We thank the reviewer for their time and effort.

Reviewer 3 Report

It's very interesting (and it would be a good subject for a future paper) the fact that there are inmates pilgrims to Santiago. They remember us the "Libri Poenitentiales" and the medieval laws.

Author Response

We really appreciate your feedback on our article, which encourages us to 
do more and better. We thank the reviewer for his time and effort.
It is indeed a very interesting subject to explore in future publications, so 
we appreciate the suggestion.

Round 2

Reviewer 1 Report

The author(s) has responded well to the feedback and I have made only minor edits to the draft.

The author(s) has responded well to the feedback and I have made only minor edits to the draft.

Author Response

Author's Reply to the Review Report - Reviewer 1 – Round 2

We appreciate the reviewer suggestions and opinions which, of course, contribute to the improvement of our article, so we have taken them into account. We thank the reviewer for the time and effort.

The reviewer was kind enough to make again the comments and suggestions directly in the text. Given the difficulty of mentioning each of them here, we have chosen again to make the changes also directly in the text.

It can be noted that all comments and suggestions have been considered, contributing to an improvement of the article.

Thank you very much.